# Different Expression of Mitochondrial and Endoplasmic Reticulum Stress Genes in Epicardial Adipose Tissue Depends on Coronary Atherosclerosis

**DOI:** 10.3390/ijms22094538

**Published:** 2021-04-26

**Authors:** Helena Kratochvílová, Miloš Mráz, Barbora J. Kasperová, Daniel Hlaváček, Jakub Mahrík, Ivana Laňková, Anna Cinkajzlová, Zdeněk Matloch, Zdeňka Lacinová, Jaroslava Trnovská, Peter Ivák, Peter Novodvorský, Ivan Netuka, Martin Haluzík

**Affiliations:** 1Centre for Experimental Medicine, Institute for Clinical and Experimental Medicine, Vídeňská 1958, 140 21 Prague 4, Czech Republic; krth@ikem.cz (H.K.); AnnaCinkajzlova@seznam.cz (A.C.); lacz@ikem.cz (Z.L.); troj@ikem.cz (J.T.); p.novodvorsky@sheffield.ac.uk (P.N.); 2First Faculty of Medicine, Institute of Medical Biochemistry and Laboratory Diagnostics, Charles University and General University Hospital, U Nemocnice 499/2, 128 08 Nové Město, Prague, Czech Republic; mrzm@ikem.cz; 3Department of Diabetes, Institute for Clinical and Experimental Medicine, Vídeňská 1958, 140 21 Prague 4, Czech Republic; kapb@ikem.cz (B.J.K.); laki@ikem.cz (I.L.); 4Department of Cardiac Surgery, Institute for Clinical and Experimental Medicine, Vídeňská 1958, 140 21 Prague 4, Czech Republic; hlad@ikem.cz (D.H.); ivap@ikem.cz (P.I.); ivne@ikem.cz (I.N.); 5Department of Anesthesia and Resuscitation, Institute for Clinical and Experimental Medicine, Vídeňská 1958, 140 21 Prague 4, Czech Republic; mawj@ikem.cz; 6Shackleton Department of Anaesthesia UHS NHS UK, Southampton General Hospital, Southampton SO14, UK; zmatloch@yahoo.com; 7Department of Oncology & Metabolism, University of Sheffield, Sheffield S0114, UK

**Keywords:** coronary artery disease, epicardial fat, mitochondrial dysfunction, endoplasmic reticulum stress, gene expression, diabetes mellitus, inflammation, cardiac surgery

## Abstract

The aim of our study was to analyze mitochondrial and endoplasmic reticulum (ER) gene expression profiles in subcutaneous (SAT) and epicardial (EAT) adipose tissue, skeletal muscle, and myocardium in patients with and without CAD undergoing elective cardiac surgery. Thirty-eight patients, 27 with (CAD group) and 11 without CAD (noCAD group), undergoing coronary artery bypass grafting and/or valvular surgery were included in the study. EAT, SAT, intercostal skeletal muscle, and right atrium tissue and blood samples were collected at the start and end of surgery; mRNA expression of selected mitochondrial and ER stress genes was assessed using qRT-PCR. The presence of CAD was associated with decreased mRNA expression of most of the investigated mitochondrial respiratory chain genes in EAT, while no such changes were seen in SAT or other tissues. In contrast, the expression of ER stress genes did not differ between the CAD and noCAD groups in almost any tissue. Cardiac surgery further augmented mitochondrial dysfunction in EAT. In our study, CAD was associated with decreased expression of mitochondrial, but not endoplasmic reticulum stress genes in EAT. These changes may contribute to the acceleration of coronary atherosclerosis.

## 1. Introduction

Cardiovascular diseases are the most common cause of death worldwide [1]. The rising prevalence of type 2 diabetes mellitus (T2DM) and obesity worldwide is significantly contributing to increased cardiovascular risk. Both disorders are closely interconnected with other diseases, such as arterial hypertension and hyperlipidemia, leading to the development of atherosclerosis and myocardial ischemia due to coronary artery disease (CAD).

A close association between CAD and the quality, volume, and overall layout of adipose tissue is well-established [2]. Yet, not all adipose tissue compartments play an equally important role in the development of CAD. It is, in particular, the epicardial adipose tissue depot, localized in the vicinity of coronary arteries and producing a vast array of endocrine- and paracrine-acting adipo-, cyto-, and chemokines and other factors, that seems to be primarily involved in coronary atherosclerosis and the development of not only CAD, but also other cardiac pathologies [3,4,5,6].

Mitochondrial metabolism is an essential component of cellular energetic homeostasis, and its derangements play an important role in a number of diseases, including obesity, T2DM, and CAD. The amount of energy in the form of adenosine triphosphate (ATP) depends on the expression of genes in each of the four main enzymatic complexes [7]. Obese patients with insulin resistance and with T2DM were found to have reduced expression of mitochondrial complexes [8]. Decreased expression of mitochondrial membrane proteins was also detected in patients with heart failure, highlighting the importance of mitochondrial function for myocardial remodeling [9].

Mitochondrial oxidative stress coexists with endoplasmic reticulum (ER) stress [10]. ER stress is an evolutionary conserved mechanism of mammalian cells closely connected with different pathologies, including inflammation, atherosclerosis, heart failure, obesity, and T2DM [11,12]. The main functions of ER are protein synthesis and folding, quality control and secretion of proteins, synthesis of fatty acids and phospholipids, assembly of lipid barriers, and metabolism of saccharides [12]. A large number of different signaling pathways from the ER are able to initiate modulations of organism homeostasis. Mutations of genes important for the correct function of ER or changes in their mRNA expression can lead to ER stress and subsequent disorders [13,14].

Whether and how ER and mitochondrial stress interact is unknown. Their occurrence in epicardial adipose tissue and potential contribution to coronary atherosclerosis remain largely unexplored. Based on the prevalence of both ER and mitochondrial dysfunction in subcutaneous and visceral adipose tissue in obesity, T2DM, and CAD, as well as the recently suggested importance of epicardial adipose tissue in coronary atherosclerosis, we hypothesized that both mitochondrial and ER stress will be present in EAT of patients with CAD. To this end, we performed a comparison of mRNA expression of selected mitochondrial and ER genes in subcutaneous and epicardial adipose tissue, skeletal muscle, and right atrial myocardium in subjects undergoing elective cardiac surgery with and without established CAD.

## 2. Results

### 2.1. Antropometric and Biochemical Characteristics

At baseline, no differences between the CAD and noCAD groups in any of anthropometric characteristics were present, including age, weight, height, BMI, and waist and hip circumference, except for greater epicardial adipose tissue thickness in patients with CAD (Table 1). Additionally, patients with CAD had a higher prevalence of T2DM, arterial hypertension, and dyslipidemia, as well as increased fasting blood glucose and HbA_1c_ compared to the noCAD group, while showing no difference in serum insulin, C-peptide, or lipid parameters (Table 1).

### 2.2. Mitochondrial and Endoplasmic Reticulum Stress Gene mRNA Expression in Adipose Tissue, Myocardium, and Intercostal Muscle

#### 2.2.1. Mitochondrial Gene mRNA Expression in SAT and EAT

Basal mRNA expression of almost all assessed mitochondrial genes (*NDUFA12, CS, SDHA, CYC1, COX4/1*) in EAT was significantly lower in the CAD compared to the noCAD group (Figure 1). At the same time, *MT-ND5* mRNA expression showed a similar tendency, but the difference did not reach statistical significance. In contrast, in SAT of CAD patients, decreased baseline mRNA expression was found only for *SDHA* (Figure 1), while the expression of other mitochondrial genes did not differ between the groups. When comparing both adipose tissue depots, only *NDUFA12* showed differences between EAT and SAT, with higher expression in EAT in the noCAD group (*p* = 0.004). Surgery tended to decrease the mRNA expression of all mitochondrial genes in EAT, regardless of CAD, with *CS, NDUFA12,* and *MT-ND5* showing significant differences. In contrast, no such pattern was present in SAT, with only *MT-ND5* expression in CAD subjects being significantly reduced post-operatively (Figure 1). None of these changes could be seen when comparing diabetic and nondiabetic subjects (data not shown).

#### 2.2.2. Mitochondrial Gene mRNA Expression in the Myocardium and Intercostal Muscle

In the right atrial myocardium, mRNA expression of *CS* was significantly lower in the CAD compared to the noCAD group (*p* = 0.036), while no other significant differences in the myocardium or ICM could be observed before or after the surgical procedure between the groups in any of the other examined mitochondrial genes (data not shown).

#### 2.2.3. Endoplasmic Reticulum Stress Gene mRNA Expression

At baseline, the expression of all assessed ER stress genes did not show any differences between the groups in any tissue, except for significantly higher mRNA expression of *DDIT3* in SAT compared to EAT in noCAD subjects (*p* = 0.023). Surgery affected only EAT, with increased expression of *HSPA5* and *ATF6* in CAD and *ATF4* in both CAD and no CAD groups (Figure 2–data for SAT and EAT and data for ICM and RA are not shown).

## 3. Discussion

Adipose tissue surrounds the heart on 80% of its surface and forms up to 20% of total heart weight [2,15]. This adipose tissue around the myocardium is divided into epicardial and pericardial adipose tissue [2]. Our study focused primarily on epicardial adipose tissue. The presented data demonstrate that mitochondrial mRNA expression in EAT is substantially and consistently lower in patients with CAD compared to subjects without CAD, while SAT and other examined tissues do not show the same results. In contrast, no change in mRNA expression of ER stress genes was observed throughout the study.

For our study, we have chosen six main mitochondrial genes (*CS, NDUFA12, MT-ND5, CYC1, COX 4/1,* and *SDHA*), which have a primary role in mitochondrial respiratory chain complexes, and thus in mitochondrial function. Lower expression in EAT in patients with compared to without CAD was demonstrated in all selected mitochondrial genes except *MT-ND5*. These findings are in line with previous data published by Nakajima et al., who showed lower mitochondrial respiratory capacity in EAT in CAD patients [16]. Here, we confirm and complement these data based on mitochondrial activity with the results from gene expression. Correct mitochondrial gene expression and function has been linked to the retention of healthy anti-inflammatory and anti-atherosclerotic phenotype of epicardial adipose tissue [17,18]. Our data thus further strengthen the suggested close relationship between EAT dysfunction and coronary atherosclerosis [2]. Interestingly, no such changes could be seen in SAT, expanding the previously reported difference in pro-inflammatory status between EAT and SAT, also to mitochondrial gene expression, and again underscoring the relevance of EAT in the development of CAD. To these results, we can also add the observed decreased mRNA expression of citrate synthase in the right atrial myocardium in the CAD group, which suggests a reduction in allover mitochondrial content. These findings might contribute to a better understanding of the interconnection between CAD and other cardiac pathologies, as mitochondrial dysfunction was shown to be associated with diseases, such as atrial fibrillation and heart failure [19,20].

Our study primarily compared subjects with and without CAD, regardless of T2DM, as diabetes was present only in 37% of CAD subjects and in none of the subjects without CAD. However, after adjusting for its presence, our data show that T2DM did not influence mRNA expression of investigated mitochondrial genes. Although data on mitochondrial function of EAT, specifically in T2DM, are virtually non-existent, this seems to be in contrast with the results of Dahlman et al., who observed a downregulation of electron transport chain genes in visceral adipose tissue of obese T2DM subjects relative to non-obese healthy subjects [21]. However, as their and our cohort differ significantly with respect to BMI, comorbidities, and other factors, no clear conclusions can be drawn from this discrepancy, and further research on larger T2DM cohorts is needed to gather more information about the relationship between mitochondrial dysfunction in EAT and T2DM.

Surprisingly, we did not detect any relevant mRNA changes in ER stress genes, either in EAT or in any other investigated tissue, regardless of the presence of CAD. It is well-known that cardiovascular disorders are associated with disturbances in ER and abnormal accumulation of misfolded proteins in cardiomyocytes [22], although the available results on the relationship of ER stress with CAD are conflicting [23]. In contrast to previous studies on mice, we did not find any upregulation of ER stress genes, which might be at least partially explained by the fact that the available murine data show the association of ER stress primarily with degenerating cardiomyocytes [24]. This together with our other data suggests that EAT-associated ER stress might not play as important of a role in the development of CAD as mitochondrial dysfunction. At this stage, we were not able to confirm the mRNA findings with data on proteins, which nevertheless remains a plan for the near future.

Cardiac surgery, usually coupled with extracorporeal circulation, comprises an intense stressor for the organism, and is associated with increased systemic and local inflammatory and metabolic responses [25]. Here, we show that, in EAT, mRNA expression of half of the six mitochondrial genes decreased substantially after cardiac surgery, regardless of the presence of CAD, while this was the case for only one gene in SAT of the CAD group. This finding further strengthens the differences between SAT and EAT with regards to mitochondrial dysfunction. Interestingly, in spite of a comparable baseline expression of ER stress genes, cardiac surgery increased the expression of three out of four genes assessed in EAT of CAD subjects, and only one gene in EAT of noCAD subjects (while having no effect in SAT of both groups), suggesting that CAD-derived EAT might still be more susceptible to ER stress than the one not associated with coronary atherosclerosis.

In summary, in our study, CAD was associated with mitochondrial dysfunction in EAT, as assessed by decreased expression of mitochondrial genes, but not with ER stress. Further studies are needed to confirm these findings in larger cohorts and to define the exact mechanisms of interaction between EAT mitochondrial dysfunction and CAD, which may lead to the identification of novel therapeutic targets and strategies in the treatment of CAD.

## 4. Materials and Methods

### 4.1. Study Subjects

This study included 38 patients undergoing elective cardiac surgery with cardiopulmonary bypass (coronary artery bypass grafting (CABG) in 13 subjects, valve replacement in 17 subjects, and combination of CABG and valve replacement in eight subjects). Patients were divided into two groups according to the presence of CAD, assessed by previous selective coronarography—11 subjects without CAD (noCAD group: three females and eight males) and 27 subjects with CAD (CAD group: five females and 22 males). Ten of the participating patients had T2DM, 33 had arterial hypertension treated with antihypertensives, and 32 had dyslipidemia treated with statins. None of the patients suffered from acute or chronic kidney disease, malignancy, thyroid disease, or acute infection. All participants signed written informed consent prior to enrollment into the study. The study was approved by the Human Ethics Review Board, Institute for Clinical and Experimental Medicine and Thomayer Hospital, Prague, Czech Republic, and was performed in accordance with the guidelines proposed in the Declaration of Helsinki (2000) of the World Medical Association. Elective cardiac surgery was performed after overnight fasting and was started between 7–8 AM in all subjects. Ten patients received infusion of dobutamine and norepinephrine perioperatively, with a maximum dose of 7 μg/kg/min and 0.2 μg/kg/min, respectively, with the treatment duration from 8 to 33 h.

### 4.2. Anthropometric Examination and Blood and Tissue Sampling

All subjects included in the study were measured and weighted, and their body mass index (BMI) was calculated one day prior to surgery. Waist and hip circumferences were measured, and their ratio was calculated. Blood samples for biochemical and hormonal measurements were taken after overnight fasting prior to initiation of anesthesia (beginning of surgery) and at the end of surgery. Blood samples were centrifuged for 10 min at 3000× *g* within 30 min after withdrawal. Serum samples were subsequently stored in aliquots at −80 °C until further analysis. The thickness of EAT was measured by transthoracic echocardiography in front of the right ventricular wall from the parasternal long axis (PLAX) view. Samples of subcutaneous (thoracic region, sternotomy site) (SAT) and epicardial (anterior interventricular sulcus or right margin of the heart) adipose tissue (EAT), intercostal muscle (ICM), and the myocardial right atrium (RA) for mRNA expression analysis were taken at the start and end of the surgery from approximately the same location in all patients. Tissue samples (50–100 mg) were collected with 1 mL of RNAlater^®^ reagent (Ambion^®^ Invitrogen, Carlsbad, CA, USA) and stored at −80 °C until further analysis.

### 4.3. Hormonal and Biochemical Assays

Serum levels of cytokines were measured by a multiplex assay MILLIPLEX MAP Human High Sensitivity T Cell Panel and Cytokine/Chemokine Magnetic Bead Panel (HSTCMAG-28SK-05, HCYTOMAG-60K-06, Merck KGaA, Darmstadt, Germany). Sensitivity for IFN-γ was 0.8 pg/mL, for IL-10 0.56 pg/mL, for IL23 3.25 pg/mL, for MIP-1α 0.94 pg/mL, for MIP-1β 0.67 pg/mL, for IL-6 0.9 pg/mL, for IL-8 0.4 pg/mL, for MCP1 1.9 pg/mL, and for TNF-α 0.7 pg/mL. The intra- and inter-assay variabilities for all assays were between 5.0 and 15.0%. C-reactive protein (CRP) levels were measured by a high sensitivity assay (Bender Med Systems, Vienna, Austria), with a sensitivity of 3 pg/mL. Routine biochemical parameters were measured at the Department of Biochemistry, Institute for Clinical and Experimental Medicine, Prague, Czech Republic by standard laboratory methods. LDL cholesterol was calculated using the Friedewald equation. Insulin levels were measured by an RIA kit (Cis Bio International, Gif-sur-Yvette, France). Sensitivity was 2.0 μIU/mL.

## 5. Quantitative Real-Time PCR

### 5.1. Determination of mRNA Expression

Samples of SAT, EAT, ICM, and RA were homogenized on a MagNA Lyser instrument with MagNA Lyser Green beads (Roche Diagnostics GmbH, Mannheim, Germany). Total RNA from homogenized tissue was extracted on a MagNA Pure instrument using a Magna Pure Compact RNA Isolation kit (tissue) (Roche Diagnostics GmbH, Mannheim, Germany). The RNA concentration was determined from absorbance at 260 nm on a NanoPhotometer (Implen, Munchen, Germany). Reverse transcription was performed using 0.25 μg of total RNA to synthesize the first strand cDNA using random primers per the instructions of the High-Capacity cDNA Reverse Transcription Kit (Applied Biosystems, Foster City, CA, USA). Gene expression of inflammatory, mitochondrial, and endoplasmic reticulum stress genes was performed on a ViiA7 Instrument (Applied Biosystems, Foster City, CA, USA) using TaqMan^®^ gene expression assays (Applied Biosystems, Foster City, CA, USA). A mix of TaqMan^®^ Universal PCR Master Mix II, NO AmpErase^®^ UNG (Applied Biosystems, Foster City, CA, USA), nuclease-free water (Fermentas Life Science, Lithuania), and specific TaqManGene expression assays (Applied Biosystems, Foster City, CA, USA) was used for the reaction. Beta-2 microglobulin (*B2M*) was used as the endogenous reference. The formula 2^−ddCt^ was used to calculate relative gene expression. The complete determination process is described in detail elsewhere [26].

### 5.2. Mitochondrial and Endoplasmic Reticulum Stress Genes

For our study, we have chosen the most abundant genes of the mitochondrial respiratory chain (Appendix A). Citrate synthase (*CS*), which is considered a marker of total mitochondria, is a tricarboxylic acid cycle enzyme that catalyzes the synthesis of citrate from oxaloacetate and acetyl coenzyme A (acetyl-CoA), and is located in the mitochondrial matrix [27,28]. Other study genes are located in the inner mitochondrial membrane, including nicotinamide adenine dinucleotide hydrogen (NADH) dehydrogenase subunit (*NDUFA12*) and mitochondrially encoded NADH dehydrogenase 5 (*MT-ND5*), which are both part of an enzymatic complex NADH-coenzymeQ_10_ reductase (Complex I), succinate dehydrogenase complex flavoprotein subunit A (*SDHA*), which encodes a major subunit of succinate-ubiquinone oxidoreductase (complex II), cytochrome c oxidase subunit 4/1 (*COX4/1*), which is part of cytochrome c oxidase (complex IV) [29], and cytochrom c 1 (*CYC1*), which is a respiratory subunit of ubiquinol cytochrome c reductase (complex III) [30] [31].

Assessed genes of ER stress include *HSPA5, DDIT3, ATF4,* and *ATF6* (Appendix A). *ATF4* is a stress-induced transcription factor and one of the master regulators of the cellular stress response that promotes adaptation of cells to a limited availability of nutrients [32]. *ATF6* is a transcription factor located in the ER membrane and sensing ER stress. Without stress conditions, ATF6 occurs as a 90 kDA trans-membrane glycoprotein (p90ATF6). Under stress conditions, the protein is cleaved to give a 50 kDA protein (p50ATF6), which will get into the cell nucleus, where it directly affects the expression of the uncoupling protein response target genes [33]. The protein encoded by the *HSPA5* gene is a member of the heat shock protein 70 (HSP70) family. It is localized in the lumen of the endoplasmic reticulum (ER) and is involved in the folding and assembly of proteins in the ER [34]. The *DDIT3*-encoded protein is implicated in adipogenesis and erythropoiesis, is activated by endoplasmic reticulum stress, and promotes apoptosis [35].

## 6. Statistical Analysis

Statistical analysis was performed, and graphs were drawn using SigmaPlot 13.0 (Systat Software Inc., San Jose, CA, USA). Results are expressed as means ± standard errors of the mean (SEM) or median (interquartile range), according to the normality of the data. Normality of all data was assessed by the Shapiro–Wilk test. An unpaired *t*-test, Mann–Whitney rank sum test, paired test, or Wilcoxon signed rank test were used for the assessment of intra- and intergroup differences, as appropriate. Correlations were analyzed using Spearman’s or Pearson’s correlation test, according to the normality of the data. Multiple linear regression analysis using a backward stepwise variable selection method was performed in the combined group of all study subjects, using parameters with significant results from Spearman or Pearson correlation tests. Values were adjusted for the presence of T2DM. In all statistical tests, *p*-values < 0.05 were considered significant.

## Figures and Tables

**Figure 1 ijms-22-04538-f001:**
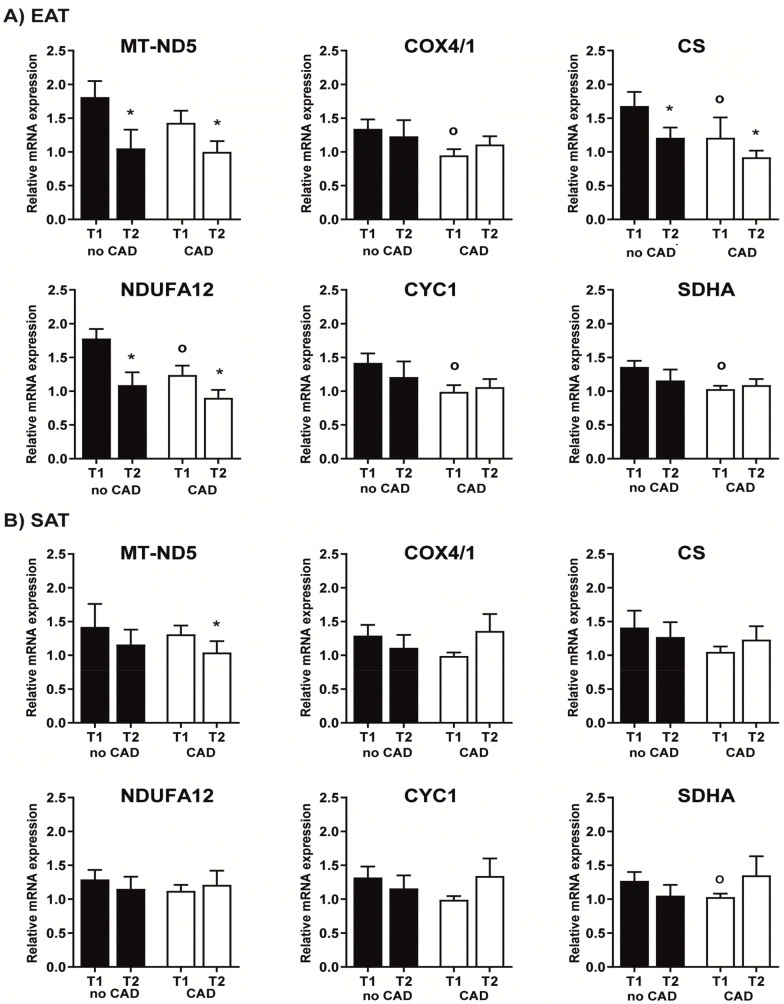
Mitochondrial gene mRNA expression in epicardial (EAT, **A**) and subcutaneous (SAT, **B**) adipose tissue. Values are shown as the mean ± SEM. Statistical significance is from an unpaired *t*-test and paired *t*-test (T1 vs. T2); * *p* < 0.05 vs. T1.; *p* < 0.05 vs. no CAD T1. Values were adjusted for the presence of type 2 diabetes mellitus. Relative mRNA expression = changes in gene expression in a given sample relative to the reference sample (beta2-microglobulin). CAD, coronary artery disease; T1, beginning of the surgery; T2, end of the surgery; CS, citrate synthase; *NDUFA12*, NADH dehydrogenase subunit; *SDHA*, succinate dehydrogenase complex flavoprotein subunit A; *COX4/1*, cytochrome c oxidase subunit 4/1; *CYC1*, cytochrom c 1; *MT-ND5*, NADH-ubiquinone oxidoreductase chain 5.

**Figure 2 ijms-22-04538-f002:**
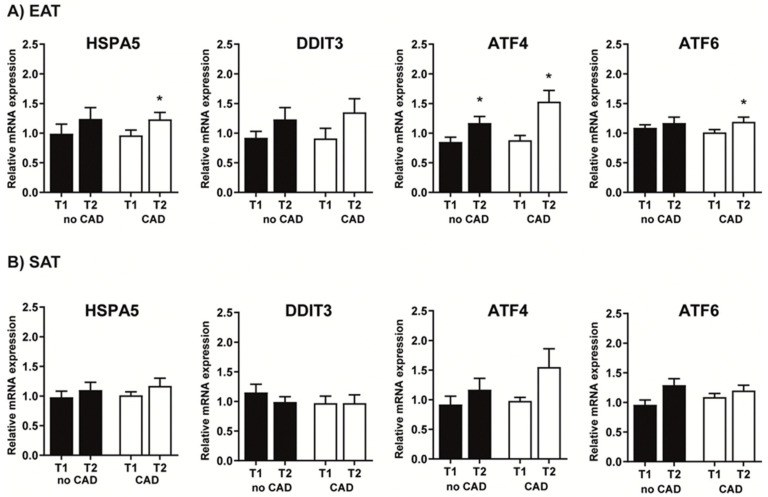
ER stress gene mRNA expression in epicardial (EAT, **A**) and subcutaneous (SAT, **B**) adipose tissue. Values are shown as the mean ± SEM. Statistical significance is from an unpaired *t*-test and paired *t*-test (T1 vs. T2); * *p* < 0.05 vs. T1; *p* < 0.05 vs. no CAD T1. Relative mRNA expression = changes in gene expression in a given sample relative to the reference sample (beta2-microglobulin). CAD, coronary artery disease; T1, beginning of the surgery; T2, end of the surgery; *HSPA5*, heat shock protein family A (Hsp70) member 5; *DDIT3*, DNA damage inducible transcript 3; *ATF4*, activating transcription factor 4; *ATF6*, activating transcription factor 6.

**Table 1 ijms-22-04538-t001:** Baseline characteristics of study subjects. Normally distributed data are shown as the mean ± SEM. Non-parametric data are shown as the median (interquartile range). Statistical significance is from an unpaired *t*-test; *p* < 0.05 vs. no CAD, statistically significant comparisons are highlighted in bold.

Group	noCAD	CAD	P
No. of subjects (f/m)	11 (3/8)	27 (5/22)	x
Age (year)	59.8 ± 4.8	67.6 ± 1.65	0.421
Weight (kg)	80.5 ± 3.98	87.7 ± 3.07	0.193
Height (cm)	172 ± 3.09	175 ± 1.55	0.348
Body mass index (kg/m^2^)	27.1 ± 0.96	28.6 ± 0.91	0.339
Waist circumference (cm)	99.2 ± 3.53	104 ± 2.42	0.332
Hip circumference (cm)	102 ± 3.08	107 ± 1.51	0.149
Waist/hip ratio	0.96 ± 0.01	0.96 ± 0.02	0.872
Epicardial adipose tissue (mm)	3.00 (2.00–3.00)	4.00 (3.00–4.00)	0.014
Total cholesterol (mmol/L)	4.12 ± 0.29	3.76 ± 0.18	0.237
Triglycerides (mmol/L)	1.33 ± 0.16	1.45 ± 0.18	1
LDL cholesterol (mmol/L)	2.36 ± 0.24	1.99 ± 0.14	0.163
HDL cholesterol (mmol/L)	1.25 ± 0.12	1.15 ± 0.09	0.126
Fasting glucose (mmol/L)	5.4 ± 0.17	6.76 ± 0.33	0.011
HbA_1c_ (mmol/mol)	35.4 ± 1.18	43.6 ± 1.98	0.012
Insulin (µIU/mL)	18.4 (14.4–20.2)	24.2 (16.7–37.5)	0.095
C-peptide (ng/mL)	2.76 ± 0.32	3.05 ± 0.22	0.497
T2DM (*n*, %)	0 (0%)	10 (37.0%)	0.021
Arterial hypertension (*n*, %)	7 (63.6%)	26 (96.3%)	0.008
Dyslipidemia (*n*, %)	6 (54.5%)	26 (96.3%)	0.002

CAD, coronary artery disease; T2DM, type 2 diabetes mellitus; f, female; m, male.

## Data Availability

All data that support the findings of this study are available from the corresponding author upon reasonable request.

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
