# Peer review of "Different Expression of Mitochondrial and Endoplasmic Reticulum Stress Genes in Epicardial Adipose Tissue Depends on Coronary Atherosclerosis"

_ijms, 2021, doi:10.3390/ijms22094538_

Round 1

Reviewer 1 Report

Firstly, I commend the authors for very impressive and extensive work. However, note the following minor corrections:

Line 29: ….included in the study.

Line 129: …. unpaired t-test and paired t-test…..

Line 145: … heart on 80% of its surface and forms up to 20% of total…

Lines 157 and 211: …. Nakajima et al. …..

Lines 212-215: I suggest the authors to end strong to showcase the significance of their work than listing the limitations of this study.

Author Response

We thank the reviewer for his/her kind review and comments. We have corrected the manuscript accordingly. Specific answers are listed below:

Line 29: ….included in the study.

Corrected accordingly

Line 129: …. unpaired t-test and paired t-test…..

Corrected accordingly

Line 145: … heart on 80% of its surface and forms up to 20% of total…

Corrected accordingly

Lines 157 and 211: …. Nakajima et al. …..

Corrected accordingly

Lines 212-215: I suggest the authors to end strong to showcase the significance of their work than listing the limitations of this study.

We thank the reviewer for this suggestion - we have adapted the text accordingly.

Reviewer 2 Report

The manuscript entitled "Different expression of mitochondrial and endoplasmic reticulum stress genes in epicardial adipose tissue depends on coronary atherosclerosis" is a very important study to show the crosstalk between mitochondrial gene expression and coronary artery disease.

The study is very significant in current scenario and the result is very important for preventive and therapeutic purpose.

 Authors need to check western blot or other method to confirm that there is no involvement of ER stress and only mitochodrial dysfunction is involved. (as they conclude in  this manuscript).  Specifically when the sample (human control/ patient) number is low, it is advisable to show another method to confirm their result/conclusion.

Author Response

We thank the reviewer for his/her kind review and comments. We fully agree that a protein analysis would further strengthen our conclusions; however due to time and material constraints we are currently unable to perform the suggested analyses. Nevertheless, we plan to expand our findings also to the protein level in the near future. We have added this information into the Discusssion section, lines 200-202.